

# Social network community structure and the contact-mediated sharing of commensal *E. coli* among captive rhesus macaques (*Macaca mulatta*)

Krishna Balasubramaniam[1], Brianne Beisner[1,2], Jiahui Guan[3],
Jessica Vandeleest[1,2], Hsieh Fushing[3], Edward Atwill[1] and Brenda McCowan[1,2]

[1] Department of Population Health & Reproduction, School of Veterinary Medicine, University of California, Davis, CA, United States of America
[2] Brain, Mind & Behavior, California National Primate Research Center, University of California, Davis, CA, United States of America
[3] Department of Statistics, University of California , Davis, CA, United States of America

Corresponding author
Krishna Balasubramaniam,
krishnanatarajan@ucdavis.edu

## ABSTRACT

In group-living animals, heterogeneity in individuals' social connections may mediate the sharing of microbial infectious agents. In this regard, the genetic relatedness of individuals' commensal gut bacterium *Escherichia coli* may be ideal to assess the potential for pathogen transmission through animal social networks. Here we use microbial phylogenetics and population genetics approaches, as well as host social network reconstruction, to assess evidence for the contact-mediated sharing of *E. coli* among three groups of captively housed rhesus macaques (*Macaca mulatta*), at multiple organizational scales. For each group, behavioral data on grooming, huddling, and aggressive interactions collected for a six-week period were used to reconstruct social network communities via the Data Cloud Geometry (DCG) clustering algorithm. Further, an *E. coli* isolate was biochemically confirmed and genotypically fingerprinted from fecal swabs collected from each macaque. Population genetics approaches revealed that Group Membership, in comparison to intrinsic attributes like age, sex, and/or matriline membership of individuals, accounted for the highest proportion of variance in *E. coli* genotypic similarity. Social network approaches revealed that such sharing was evident at the community-level rather than the dyadic level. Specifically, although we found no links between dyadic *E. coli* similarity and social contact frequencies, similarity was significantly greater among macaques within the same social network communities compared to those across different communities. Moreover, tests for one of our study-groups confirmed that *E. coli* isolated from macaque rectal swabs were more genotypically similar to each other than they were to isolates from environmentally deposited feces. In summary, our results suggest that among frequently interacting, spatially constrained macaques with complex social relationships, microbial sharing via fecal-oral, social contact-mediated routes may depend on both individuals' direct connections and on secondary network pathways that define community structure. They lend support to the hypothesis that social network communities may act as bottlenecks to contain the spread of infectious agents, thereby encouraging disease control strategies to focus on multiple organizational scales. Future directions include

increasing microbial sampling effort per individual to better-detect dyadic transmission events, and assessments of the co-evolutionary links between sociality, infectious agent risk, and host immune function.

# INTRODUCTION

In social systems, understanding the dynamics of infectious agent transmission among individuals remains critical for the development of disease control strategies (*Alexander, 1974*; *Drewe & Perkins, 2015*; *McCowan et al., 2016*; *Nunn, 2012*; *Schmid-Hempel, 2017*). Across a wide range of taxa, epidemiological studies have revealed strong links between the prevalence and diversity of infectious agents, and exposure to contact-based social interactions among individuals (reviewed in *Drewe & Perkins, 2015*; *Rushmore, Bisanzio & Gillespie, 2017*; *VanderWaal & Ezenwa, 2016*). Yet infectious agent acquisition may be impacted by phenomena other than contact-based sharing, for instance host physiological characteristics like stress- or immune-responses (*Cohen, Janicki-Deverts & Miller, 2007*; *Sapolsky, Romero & Munck, 2000*; *Segerstrom & Miller, 2004*), and/or the presence of strong connections, via mitigating stress-levels and/or enhancing immune function, socially buffering individuals against infection risk (*Balasubramaniam et al., 2016*; *Kaplan et al., 1991*; *Uchino, 2004*; *Uchino, 2009*; *Young et al., 2014*). One way to assess the clear effects of contact-mediated microbial sharing in the absence of the above phenomena is by characterizing the phylogenetic relationships of commensal (rather than pathogenic) gut-microbes (*VanderWaal & Ezenwa, 2016*). Here we use the diverse phylogenetic relationships of such a microbe, gut *Escherichia coli* (or *E. coli*), along with social network analyses, to assess social contact-mediated microbial sharing among captive rhesus macaques (*Macaca mulatta*) at multiple scales of social organization.

Among group-living animals, heterogeneity in individuals' interaction with their natural and/or social environment may strongly influence their exposure to infectious agents. Social network analyses, which incorporate both direct and secondary pathways of contact, have revealed that central individuals, and those with higher numbers and strengths of both primary and secondary connections in their social network have (i) higher endoparasite loads, (ii) greater prevalence of a specific pathogen, and/or (iii) higher pathogenic diversity (e.g., bumble bees: *Otterstatter & Thomson, 2007*; group-living lizards: *Godfrey et al., 2009*; Tasmanian devils: *Hamede et al., 2009*; meerkats: *Drewe, 2010*; Belding's ground-squirrels: *VanderWaal et al., 2013a*; nonhuman primates: *Balasubramaniam et al., 2016*; *MacIntosh et al., 2012*; *Rimbach et al., 2015*). Yet without an assessment of microbial similarity, such studies can only indirectly infer whether microbial sharing or transmission *might* occur via social and/or space-use networks, rather than confirm that transmission *did* occur.

Comparison of microbial genetic data from hosts can help resolve the dynamics of infectious agent transmission (*VanderWaal & Ezenwa, 2016*). Commensal gut microbes

are ideal models for detecting the potential for contact-mediated pathogen transmission at a high resolution by virtue of being present in almost every individual in a group. The sharing of commensal microbes is not affected by alternative phenomena like social buffering, i.e., the investment on social capital by individuals that maybe expected to reduce their susceptibility-mediated exposure to pathogens (*Young et al., 2014*; *Balasubramaniam et al., 2016*). Further, they rarely (if ever) alter the behavior of the host (*VanderWaal et al., 2014a*), allowing researchers to study subtle sharing or transmission events that may precede the potential outbreak of an infection. Specifically, commensal *E. coli* are facultative, anaerobic, non-pathogenic bacteria that are highly prevalent in the gastrointestinal tracts of mammals (*Sears, Brownlee & Uchiyama, 1950*; *Sears et al., 1956*; *Tenaillon et al., 2010*). They exhibit a clonal population structure that is little affected by horizontal gene transfer and/or mutation within relatively shorter-term, epidemiological time-scales (*Tenaillon et al., 2010*). The genetic diversity of *E. coli* is sufficient to capture inter-individual variation in genetic profiles (*Craft, 2015*). Healthy individuals tend to carry one predominant, permanent strain of *E. coli*, and one or more (up to 13) transient strains (*Caugant, Levin & Selander, 1981*; *Sears, Brownlee & Uchiyama, 1950*; *Sears et al., 1956*). Thus if individuals have genotypically similar or identical *E. coli*, they are likely to have either shared the strain via fecal-oral contact, or through using a common environmental source (*Chiyo et al., 2014*; *Springer et al., 2016*; *VanderWaal et al., 2013b*; *VanderWaal et al., 2014a*). Finally, they may be easily isolated and characterized (*Dombek et al., 2000*; *Goldberg, Gillespie & Singer, 2006*), making them well-suited for genetic subtyping and phylogenetic tree reconstruction to infer fecal-oral sharing or transmission events.

Previous studies implementing population genetics-based approaches have revealed that *E. coli* subtype sharing commonly occurs between humans and livestock (*Goldberg et al., 2008*; *Rwego et al., 2007*), pets (*Johnson, Clabots & Kuskowski, 2008*; *Johnson et al., 2009*), and wild great apes in areas of shared space use (gorillas: *Rwego et al., 2007*; chimpanzees: *Goldberg et al., 2007*). Yet these studies, on account of not having used network-based approaches to reveal potential transmission routes, advise caution in interpreting broad similarities as evidence for contact-mediated bacterial transmission. More recently, *E. coli* sharing has been assessed among some free-living animal social groups by comparing social behavioral and space-use networks to "transmission networks". In these, transmission links among individuals were inferred based on the degrees of bacterial phylogenetic similarity (reticulated giraffes (*Giraffa camelopardalis*): *VanderWaal et al., 2013b*; African elephants (*Loxodonta africana*): *Chiyo et al., 2014*; Verreaux's Sifakas (*Propithecus verreauxi*): *Springer et al., 2016*). Here we use the term "sharing" rather than "transmission", since phylogenetic inferences at a single (or a few scattered) time-point are insufficient to determine the identities of donors and recipients in a transmission event. As in older studies, we first use a population-genetics approach to first establish a premise for expecting social contact-mediated sharing of commensal *E. coli* among groups of rhesus macaques. Further, we use network-based approaches to examine whether within groups, *E. coli* is more likely to be shared among frequently interacting dyads.

In addition to their primary connections, individuals may also acquire infectious agents via potential transmission events from less frequent, less-captured contact events

with secondary partners (*Griffin & Nunn, 2012*; *MacIntosh et al., 2012*; *Nunn et al., 2015*). The extent to which individuals prefer to interact more with specific subsets of partners may culminate in the formation of clusters, or social network communities, in some societies (*Fushing et al., 2013*; *Newman, 2006*; *Whitehead & Dufault, 1999*). In other words, microbial sharing may be discernible at higher levels of spatial or social structure in addition to, or instead of at the dyadic level. Indeed, analyses of both natural datasets and mathematical models have revealed that socially transmitted infectious agents may spread faster among individuals within the same social community or sub-group, compared to individuals across communities (*Griffin & Nunn, 2012*; *Nunn et al., 2011*; *Nunn et al., 2015*; *Huang & Li, 2007*; *Salathe & Jones, 2010*). Analogous to this ''social bottleneck hypothesis'' (*Nunn et al., 2015*), we might expect greater commensal microbial sharing among individuals of the same social behavioral communities, compared to individuals across communities. Reconstructing the social network community structures of macaque groups, we build upon previous approaches that have focused on individual or dyadic interactions, by also comparing the extent of bacterial phylogenetic similarity observed within versus across social network communities.

Rhesus macaques are an ideal host species to study the behavioral bases of bacterial sharing. They are biologically, socially and cognitively analogous to human societies (*Cobb, 1976*; *Suomi, 2011*). In nature, they live in large (approximate range: 20–150 individuals), multi-male-multi-female social groups, in which individuals maintain and reinforce their social relationships using a variety of behaviors (*Southwick & Siddiqi, 2011*; *Thierry, 2007*), For instance, allogrooming (hereafter grooming) and social huddling are the most common contact-mediated affiliative exchanges in primates (*Henzi & Barrett, 1999*). Aggressive interactions, which form the basis of rhesus group social structure (*Lindburg, 1971*; *Sade, 1972*), are also significant from an epidemiological perspective, since individuals may come into physical contact during aggressive encounters. Contact behaviors facilitate fecal-oral infectious agent transmission (e.g., grooming for helminthal transmission in Japanese macaques (*M. fuscata*) and brown spider monkeys (*Ateles hybridus*): *MacIntosh et al., 2012*; *Rimbach et al., 2015*; huddling for enteric bacterial transmission in captive rhesus macaques: (*Balasubramaniam et al., 2016*); aggression in the spread of *Mycobacterium tuberculosis* smong meerkats (*Suricata suricatta*): *Drewe, 2010*). Such findings encourage assessing the behavioral bases for the contact-mediated sharing of gut *E. coli*. Finally, rhesus societies typically show despotic, nepotistic social styles with strong tendencies for sub-grouping within their social networks (*Sueur et al., 2011b*; *Thierry, 2007*). They are hence well-suited for examining the network-mediated bases of microbial sharing at community-wide scales.

For three groups of captive rhesus macaques housed in separate enclosures, we examined evidence for the social contact-based sharing of commensal *E. coli*. We first use a population genetics approach to establish a premise for expecting contact-mediated sharing of *E. coli* within rhesus macaque groups. Specifically, we tested whether across all three macaque groups, the overall phylogenetic similarity of *E. coli* was more strongly influenced by social group membership relative to intrinsic factors like age, sex, and/or matrilineal genetic relatedness of individuals. Further, we also assessed whether the overall genetic similarity

Table 1 Demographic characteristics of the three study-groups of rhesus macaques,

| Group ID | Number of matrilines | Age (mean ± SD) | Max. age | Min. age | Year of formation | Number of adults sampled | Number of *E. coli* isolates |
|---|---|---|---|---|---|---|---|
| Group I | 13 | 8.02 ± 5.39 | 29 | 3 | 1991 | 101 | 79 |
| Group II | 13 | 8.30 ± 4.69 | 21 | 3 | 1995 | 96 | 78 |
| Group III | 26[a] | 5.94 ± 2.54 | 11 | 3 | 2005 | 102 | 86 |
| | | | | | Total | 299 | 243 |

**Notes.**
[a] Fragmented matriline structure, since the group was composed of younger individuals introduced from multiple other groups.

of *E. coli* was different across the three groups. Second, we examined whether within each macaque group, the degree of pairwise *E. coli* similarity was positively related to the frequencies of dyadic grooming, huddling, and/or aggressive interactions in their social networks. Finally, we examined whether contact-based sharing of *E. coli* was discernible at the community level. Specifically, we asked whether within each macaque group, clusters of individuals that were more connected to each other as part of the same social behavioral community were also more similar to each other in their *E. coli* subtypes, than they were to macaques in different communities.

## METHODS

### Study location and subjects

The study was conducted at the California National Primate Research Center (CNPRC) and the School of Veterinary Medicine (SVM), University of California at Davis. Data were collected on 299 adult rhesus macaques (90 males, 209 females) between 3–29 years of age (mean = 7.7 years), across three social groups (Table 1). The groups were housed in separate, 0.2 ha outdoor enclosures. Animals were fed a standard diet of monkey chow twice per day at approximately 0,700 h and between 1,430 and 1,530 h. They were provided fresh fruit or vegetables once a week, with seed-mixture being provided daily. Water was available *ad libitum*, sporadically as natural puddles but mostly via artificial sources such as taps. For more information regarding the study groups, and intergroup differences in sociodemographic characteristics, see Table 1 or *Balasubramaniam et al. (2016)*. The protocols used for this research were approved by the UC Davis Institutional Animal Care and Use Committee (IACUC; Protocol #: 18525; Office of Laboratory Animal Welfare (OLAW) Assurance Number: A3433-01), and were in accordance with the legal requirements of the jurisdictions in which the research was conducted.

### Behavioral data collection

Behavioral and biological data were collected during a 6-week sampling period per group, with two groups being observed in the spring (Group I: March–April 2013; Group III: March–April 2014) and one being was observed in the fall (Group II: September–October 2014). For each group, three observers collected data for 6 h on 4 days per week from 0,900–1,200 h and 1,300–1,600 h. Observers used an Event Sampling design to record both mild and severe aggressive interactions, and Scan Sampling to record affiliative

grooming and huddling interactions (*Altmann, 1974*). Further details regarding the precise definitions of behaviors and sub-categories (for aggression) may be found in *Balasubramaniam et al. (2016)*.

The event sampling approach has been previously proposed as being useful to optimize reliable data collection in large social groups to improve statistical power, and navigate non-independence issues that may affect the computation of social network measures (*Balasubramaniam et al., 2016*; *Farine & Whitehead, 2015*; *McCowan et al., 2011*; *Vandeleest et al., 2016*). Further, our frequency of scan sampling of affiliative interactions—once every 20 min during a six-hour duration of sampling per day—was also intense, generating approximately 432 scans in total. This scan sampling regime was sufficient to generate biologically meaningful social networks; a recent study on wild Japanese macaques revealed that for a given duration, frequent instantaneous scan sampling generates identical amounts of grooming data to focal sampling (*Romano et al., 2016*). Nonetheless, to verify that our sampling effort was adequate, we first computed the values of three social network measures—specifically Newman's eigenvector-based modularity, mean degree, and network density (see Table S1 for definitions and R packages used)—computed from each social network used in the analyses. We then compared these observed measures to the corresponding measures calculated from 1,000 permuted networks generated by bootstrapping increasingly smaller subsets (100%–10%, with decrements of 10%) of edges from the original network (*Croft et al., 2011*; *Farine & Whitehead, 2015*; *Lusseau, Whitehead & Gero, 2008*). Plots of network measures vs. percentage sampling effort revealed asymptotic trends, indicating that sufficient sampling effort had been reached (e.g., Newman's modularity: Fig. 1A–1C; Table S2).

## Bacterial isolations and DNA fingerprinting

All individuals within a particular group were sampled on the same day; this was critical to ensure comparability, since there could be significant turnover of *E. coli* genotypes in the mammalian gut (*Anderson, Whitlock & Harwood, 2006*). Further, sampling was conducted on a pre-selected day on the final week of the behavioral observation phase, in order to facilitate the detection of contact-based sharing of *E. coli* attributable to the animals' recent history of interactions. Prior to fecal collection, each animal was immobilized (10 mg/kg of ketamine) and given standard physical examinations by veterinary staff (e.g., checked for injuries, weighed). Two fresh fecal swabs were collected from every macaque at the end of the behavioral observation period, following previously published methods (*Good, May & Kawatomari, 1969*). A sterile cotton-tip swab was inserted into the rectum of each individual, rotated gently to collect fecal material, and immediately immersed into a 15 ml test-tube (labeled with the animal ID containing a sterile growth medium (Group I: Phosphate Buffer Saline (PBS); Groups II and III: Tryptic Soy Broth (TSB; BD, Franklin Lakes, NJ, USA))); a duplicate sample was taken using another sterile swab and placed into a second tube. The samples were incubated within 4 h of collection, with orbital rotation of 100 rpm at (1) 25 °C for 2 h, (2) 42 °C for 8 h, and (3) held static at 6 °C overnight. Commensal *E. coli* was isolated from the TSB enrichment. First, 10 uL of the enrichment was streaked for isolation onto MacConkey agar plates (BD, Franklin Lakes, NJ, USA) and

incubated at 37 °C for 18–24 h. From these, suspect colonies were streaked on to Eosine Methylene Blue agar (EMB) (BD, Franklin Lakes, NJ, USA) and incubated under similar conditions for a 24-hour period. Following an additional cycle of isolation, streaking and incubation on MacConkey plates, all suspect *E. coli* isolates were biochemically confirmed using Triple Sugar Iron (TSI) (Remel, Lenexa, KS, USA), Citrate (Remel, Lenexa, KS, USA), and Urea (BD, Franklin Lakes, NJ, USA), Methyl Red-Voges-Proskauer (MR-VP) (BD, Franklin Lakes, NJ, USA), and Indole (BD, Franklin Lakes, NJ, USA). These tests confirmed commensal *E. coli* from the majority of individuals sampled within each group (Table 1). Remaining individuals were deemed 'untypable', and were excluded from further analyses. Confirmed isolates were then banked and frozen within a −80 °C freezer for subsequent bacterial fingerprinting and phylogenetic reconstruction.

We used PFGE (PulseNet Pulsed Field Gel Electrophoresis, using the CDC protocol) to generate DNA fingerprint profiles for symbiotic *E. coli.* This technique is a well-established and valid method for a surface comparison across isolates (*Cesaris et al., 2007*; *Kilonzo et al., 2011*; *Kondo et al., 2010*; *Ribot et al., 2006*). While more novel techniques, such as whole genome sequencing, can be used to reconstruct the phylogenetic relationships of organisms using nucleotide datasets, we preferred PFGE owing to its sufficiency in assessing genus-typical bacterial diversity. Further, given (a) the large number of samples (299 macaques), and (b) our focus on a specific, non-pathogenic inhabitant of the gut microbiome (commensal *E. coli*), implementing metagenomic processing to develop complete profiles of gut microbiota was beyond the scope of this study. Finally, PFGE has been shown to perform well in previous research that links microbial sharing with the spatial and social contact networks of African ungulates (*VanderWaal et al., 2013a*; *VanderWaal et al., 2014a*).

From a single banked isolate from each individual macaque, we streaked bacterial colonies onto Tryptic Soy Agar (TSA), which in turn facilitated the production of agarose plugs containing the lysed bacterial colonies. Following this, bacterial DNA was digested with Xba-1 restriction enzyme, loaded and run through an agarose gel, and stained with ethidium bromide to visualize DNA banding patterns. DNA fingerprints were then grouped and standardized using the Bionumerics software (version 6.6, Applied Maths, Inc). Following this, we used a bandmatching procedure to extract a binary, bipartite matrix of bacterial genotypes (Table S3). Rows represented individual monkeys, and the columns each of 66 band positions identified and optimized using automated analytical parameters to fit the dataset. Cells in this matrix indicated either a presence (1) or absence (0) of a band for each individual in each band position. We also reconstructed three phylogenetic trees of bacterial similarity, one for each of the three study groups (e.g., Group I: Fig. 1). For this, we used the UPGMA (Unweighted Paired Group Method with Arithmetic-mean) procedure, which provides reliable topologies via bandmatching of fingerprint data. From each group-specific tree, we extracted a "similarity" matrix of pairwise cophenetic coefficients, i.e., the correlation between the similarity in densitometric curves and the phylogenetic branch length distance between each pair of macaques.

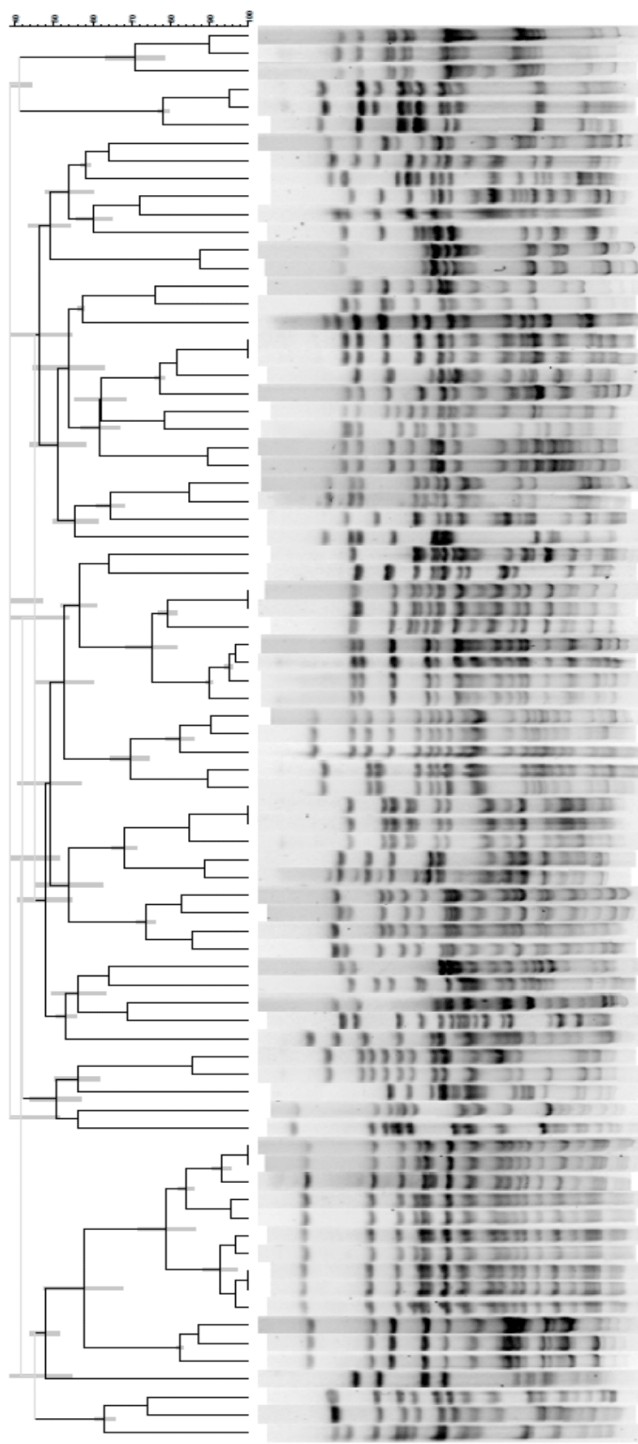

**Figure 1** **Phylogenetic tree of *E. coli* genotypic relationships isolated from 79 rhesus macaques in group I.** The tree was constructed using the UPGMA algorithm in the software Bionumerics (*version 6.6, Applied Maths Inc.*).

## Bacterial isolations from environmentally deposited feces

Our study system and design were particularly well-suited for examining evidence for microbial sharing via socially-mediated fecal-oral contact-routes. First, there was little variation in climatic conditions at CNPRC within and across the observation periods. The monkeys experienced generally hot and dry conditions, with extended periods of sunlight. Although some strains of *E. coli* may generally persist in the environment for between eight-25 weeks, hot and dry environmental conditions are considered highly unfavorable for the survival of gut *E. coli* deposited into the environment (*Habteselassie et al., 2008*; *Sinton, Hall & Braithwaite, 2007*; *Van Elsas et al., 2011*). Further, the water supply from the artificial taps tested negative for commensal *E. coli*. For these reasons, the sharing of *E. coli* on account of environmental space-use overlap, and/or from using the same artificial water-sources was highly unlikely. Nonetheless, to rule out *E. coli* sharing on account of independent acquisition from environmental feces, we compared, for one of our study groups, the *E. coli* strains isolated and fingerprinted from macaque rectal swabs to isolates from environmentally deposited feces. Specifically, we collected 18 environmental fecal samples, three from within each of six, equally divided sections of the enclosure, on the same day of macaque rectal swab collection. To collect each sample, we used a sterile sponge held with a pair of forceps to wipe a roughly 0.5 m ×0.5 m area on a man-made enrichment surface (e.g., perching frames, rails) that contained freshly deposited macaque feces. The sponges were then immediately immersed in sterile, pre-labeled bags containing TSB, which were then incubated and processed simultaneously with the macaque rectal swabs for *E. coli* confirmation and fingerprinting, using the procedures described above.

## Population genetics, social network, and statistical analyses

To examine whether the overall population genetic diversity of *E. coli* was influenced by group membership controlling for other individual attributes, we performed a series of Analysis of Molecular Variance (AMOVA: *Excoffier, Smouse & Quattro, 1992*) tests with 1,000 random permutations of the data, using the *pegas* package in R (*Paradis, 2010*). In each test, the "outcome" was a square matrix of Euclidean distances of *E. coli* genetic similarity between each pair of individuals, calculated from the bipartite matrix of band presence-absence extracted using the bandmatching procedure (Table S3). On these matrices, we ran three hierarchical or nested AMOVAs. With group membership (I, II, and III) used as a level-two variable, we ran models with (T1) species-typical age-class (old (>13 years), prime (between 4 and 13 years), (T2) sex-class (males, females), and (T3) matriline membership (individuals within the same matriline were grouped together) each nested within groups. This last analysis was run for just groups I and II, since the matriline structure for Group III was highly fragmented on account of this group being primarily composed of younger individuals introduced from various other groups (Table 1; *Balasubramaniam et al., 2016*). To examine whether the overall *E. coli* genetic similarity among individuals was different across the three groups, we first calculated the row-wise mean % similarity of each individual's *E. coli* genotype with its group members, from the similarity matrices extracted from the group-specific phylogenetic trees. We then ran a one-way ANOVA with a Tukey HSD posthoc test for multiple comparisons, with

individuals' mean similarity coefficient set as the continuous variable, and "group ID" set as the fixed factor (Levene's homogeneity of variance test: $F = 2.12$, $df = 2$, $p = 0.12$).

To examine whether the frequencies of dyadic social behavioral interactions influenced the % similarity in *E. coli* genotypes, we ran multiple, univariate Multiple Regression Quadratic Assignment Procedure (or MR-QAP) models with double dekker semi-partialling and 1,000 permutations (*Dekker, Krackhardt & Snijders, 2007*; *Hubert, 1987*; *Krackhardt, 1987*). MR-QAP accounts for the non-independence of dyadic datasets (*Hanneman & Riddle, 2005*) by coercing matrices into vectors. After performing a standard linear regression across the corresponding cells of a dependent matrix and one or more independent or co-variate matrices, the procedure uses a Monte Carlo method to randomly permute the rows and columns of the dependent matrix. It thus re-computes regression coefficients 1,000 times to generate a distribution of coefficients against which the observed coefficients may be compared. The semi-partialing approach has been shown to be fairly robust to the distribution (normal, gamma, negative binomial) of values in the outcome matrices (*Dekker, Krackhardt & Snijders, 2007*). Thus, although the distribution of *E. coli* % similarity deviated significantly from normality for all three groups (e.g., Group I: Shapiro–Wilcoxon test: $w = 0.96$, $p < 0.01$), we still ran and interpreted linear MR-QAP matrix regressions. For each macaque group, we ran four univariate models. In each model, the dependent network was the "similarity" matrix of cophenetic correlation coefficient matrix of % similarity in *E. coli*. We used the *netlm* linear function in the SNA R package (*Butts, 2008*). Independent networks included frequencies of (m1) grooming, (m2) huddling, and (m3) aggression. We included both mild and severe aggression in the aggression networks; although mild aggression does not involve direct contact, it was included since it may be linked to the likelihood of occurrence of moderate and severe aggression in captively housed macaques that interact frequently (*Balasubramaniam et al., 2016*). Since our social networks were not collinear (range of Pearson's ($r$) row-wise matrix correlation coefficients for all pairs of networks and all groups: $0.02 < r < 0.68$), we also ran a single multivariate model per group that included all three types of networks as predictors of *E. coli* similarity (Table S4). Previous studies have suggested that a major component of nonhuman primate gut microbiota may be evolutionarily conserved, or inherited (*Ley et al., 2008*; *McCord et al., 2014*). Further, social interactions among macaques may be more likely to occur among closely related maternal kin (*Berman, 2011*; *Chapais, 2006*). For these reasons, we also regressed a (m4) binary matrix of kinship (1: close kin dyads with a relationship coefficient ($r$) of $\geq 0.125$; 0: distant kin or unrelated individuals ($r < 0.125$)), on *E. coli* genotypic similarity. We ran the kinship analyses only for Groups I and II since the matriline structure for Group III was highly fragmented (see above).

Despite its utility in handling interdependencies in the data, the MR-QAP method has several restrictions. In contrast to the ordinary least squares method, it is not possible to calculate degrees of freedom, statistical power, or effect sizes in MR-QAP regression (*Ferrin, Dirks & Shah, 2006*). R-squared values also tend to have little meaning (*Gibbons, 2004*; *Zagenczyk et al., 2013*). Further, there is controversy in the use of goodness-of-fit statistics (e.g., AIC) for MR-QAP. So rather than likelihood-based model selection criteria,

we interpreted all results using the β coefficients, and the *p* values computed based on permutation tests which is the primary statistic of interest in MR-QAP analyses (*Zagenczyk et al., 2013*).

To determine whether *E. coli* sharing is more readily detectable across sets of closely interacting or spatially associated individuals, we used the Data Cloud Geometry (or DCG) method to reconstruct behavioral community structures (*Fushing & McAssey, 2010*; *Fushing et al., 2013*; *McCowan et al., 2016*). DCG identifies network community structure at multiple levels by performing a random walk through an empirical network guided by the data. Cumulatively, these random walks produce a similarity matrix describing the pairwise similarity in social connections, from which a hierarchical tree of clustering is generated (*Fushing & McAssey, 2010*; *Fushing et al., 2013*). From a biological perspective, a DCG cluster may be therefore defined as a subset of group members whose social ties are both closer to, and stronger among each other, than they are to other group members (*VanderWaal et al., 2014b*). Such "closer" individuals, whether close kin (*Berman, 2011*), non-kin allies (*Seil et al., 2017*), and/or strong social bond investors (*Silk, Alberts & Altmann, 2003*), tend to cluster together at a lower level of this tree than individuals with fewer connections and/or similarities in connections. We used DCG because it offers specific advantages over other commonly used methods to identify cluster structure, such as Hierarchical Clustering (*Corpet, 1988*; *Johnson, 1967*) (summarized in *Fushing et al., 2013*; *VanderWaal et al., 2014b*). First, it does not require dyadic relationships to be binary and instead, utilizes the strength of relationships (e.g., frequencies of behavioral interactions). Second, in comparison with Hierarchical Clustering trees, DCG trees are more robust, less sensitive to measurement errors, and provide information on the intrinsic scales embedded within the data cloud. This is because of the implementation of stricter rules for assigning nodes to the same cluster (ultrametric or strong triangle inequality rule), making DCG more accurate in identifying cluster structure (*Fushing et al., 2013*; *VanderWaal et al., 2014b*).

We constructed three DCG trees per macaque group, one each from dyadic grooming, huddling, and aggression. We used Monte Carlo tests to determine whether the observed cluster membership of macaques was significant. At each hierarchical level for each tree, we generated 1,000 random clustering configurations. We then compared the observed mean behavioral frequency within a cluster, to a distribution of mean frequencies of behaviors from the randomly generated cluster configurations. We thus considered a particular level to be significant if this mean fell within the 95th percentile of the permuted distribution (as in *VanderWaal et al., 2014b*). To determine whether % similarity in *E. coli* was greater among individuals within the same, versus across different DCG communities, we used a series of Wilcoxon rank-sum tests. We favored a non-parametric test since the dyadic *E. coli* similarity coefficients were deviated significantly from normality (see above). Nevertheless, we also ran a series of randomization tests, which compared the observed mean within-cluster *E. coli* similarity coefficients to a distribution of coefficients generated from 1,000 permuted datasets in which individuals' behavioral cluster membership was assigned randomly. Finally, to rule out *E. coli* sharing on account of macaques' shared exposure to environmental feces, we used an additional Wilcoxon rank-sum test to compare the mean % similarity of macaque-macaque *E. coli* isolates to the % similarity of
**Table 2  Hierarchical or Nested Analyses of Molecular Variance (AMOVAs) testing for the effect of group membership on the variance of *E. coli* genotypic diversity across three groups of captive rhesus macaques (244 isolates in total).** *P* values indicate significance based on randomization tests after 1,000 permutations.

|    | Source of variation | SSD | MSD | Sigma$^2$ | %Sigma$^2$ | df | p($\alpha = 0.01$) |
|----|---------------------|-----|-----|-----------|------------|-----|---------------------|
| T1 | Age nested within Groups | | | | | | |
|    | Among Groups | 60.10 | 30.05 | 0.24 | 93.63 | 2 | 0.001** |
|    | Among age categories | 53.00 | 10.60 | 0.01 | 6.37 | 5 | 0.996 |
| T2 | Sex nested within Groups | | | | | | |
|    | Among groups | 60.10 | 30.05 | 0.20 | 73.72 | 2 | 0.001** |
|    | Among sex categories | 38.06 | 12.69 | 0.07 | 26.28 | 3 | 0.061 |
| T3 | Matriline nested within Groups (Groups I & II only) | | | | | | |
|    | Among Groups | 29.45 | 29.45 | 0.26 | 75.00 | 1 | 0.012* |
|    | Among matrilines within Groups | 22.55 | 9.67 | 0.09 | 25.00 | 23 | 0.87 |

**Notes.**
** $p < 0.01$.
* $p < 0.05$.
SSD, Sum of Squares Deviation; MSD, Mean Squared Deviation; Sigma, Observed variance in genotypic diversity.

macaque-environmental fecal isolates for Group II. All statistical analyses were performed using R (ver 3.1.3), with the value of $\alpha$ being set after a Bonferroni correction for some relevant tests (AMOVA tests: $\alpha = 0.02$; MR-QAP tests: $\alpha = 0.01$; Wilcoxon rank-sum tests: $\alpha = 0.02$).

## RESULTS

### Intergroup variation in bacterial genotypic diversity

From 299 individual macaques, we isolated, confirmed, and generated a fingerprint profile for 243 individuals, 79 in Group I, 78 in Group II, and 86 in Group III. Table 2 shows the results from the AMOVA tests. As predicted, group membership was responsible for the highest proportion of the observed genotypic variance, despite the nesting of age-category, sex-category, and matriline membership within groups (Table 2). In comparison to group membership, these nested variables had little effect on *E. coli* genetic variance. Further, permutation tests associated with the AMOVAs showed that the genetic variance across different groups was significantly greater than chance datasets in which group membership was assigned randomly to the isolates. In contrast, the variance across age-categories, sex-categories and matrilines nested within groups failed to reach significance.

We found significant differences in the mean % similarity of *E. coli* of individuals across groups (one-way ANOVA: $F_{2,243} = 52.8$, $p < 0.01$; Fig. 2). A post-hoc Tukey test revealed that the mean similarity was significantly higher among the macaques of Group III ($M = 50.31$, $SD = 4.33$), in comparison to macaques in Groups I ($M = 45.39$, $SD = 5.08$), and II ($M = 43.53$, $SD = 3.68$) (Fig. 2).

### Effects of dyadic social behavioral interactions and kinship on *E. coli* similarity

Contrary to our predictions, MR-QAP models showed no clear associations between pairwise *E. coli* % similarity coefficients and the dyadic frequencies of social behavioral

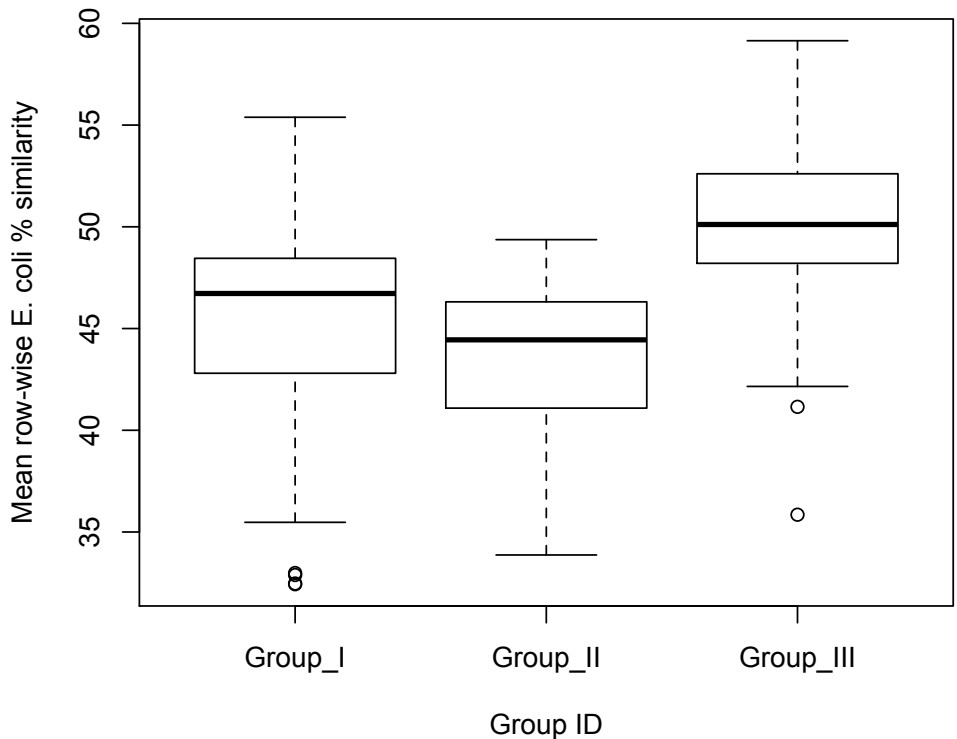

**Figure 2  Box-plot indicating the effect of group membership on *E. coli* % similarity.** Mean similarity coefficient was significantly greater among macaques in group III in comparison to those in Groups I and II.

interactions, and/or kinship (Table 3). Such a lack of association was largely consistent across all three types of contact—grooming, huddling, and aggression—frequencies. Although huddling and aggression frequencies both showed a positive impact on *E. coli* similarity in Group II, the effects were not significant after correcting for multiple comparisons (Table 3). The directions of coefficients were highly inconsistent across the co-variate matrices. For instance, although both huddling and aggression appeared to have positive associations with *E. coli* similarity for Group II, grooming showed a negative coefficient. Further, *E. coli* similarity was also not higher among close-kin dyads. Finally, such inconsistencies between dyadic social behaviors and *E. coli* % similarity were also a feature of the results from multivariate models that combined all three types of social networks (Table S4).

## Bacterial sharing among macaque social network communities

Reconstructions of social network community structures using the DCG approach revealed multiple hierarchical levels of clustering, in which individuals were embedded in higher-order communities and sub-communities. The characteristics of the DCG tree for the different study groups and types of behavioral networks are summarized in Table 4. Figure 3 shows the trees for Group I. We identified between three and five hierarchical levels of clustering in all the trees. Permutation tests run on each tree and at each hierarchical level

**Table 3** Univariate MR-QAP regression models examining the effects of dyadic social behavioral interactions and kinship on the % genetic similarity of *E. coli.*

| Model | Group I | | Group II | | Group III[a] | |
|---|---|---|---|---|---|---|
| | *B* | $p(\alpha = 0.01)$ | *B* | $p(\alpha = 0.01)$ | *B* | $p(\alpha = 0.01)$ |
| *E. coli* % similarity $\sim$ Grooming freq. | −5.91 | 0.24 | −3.36 | 0.4 | −4.1 | 0.32 |
| *E. coli* % similarity $\sim$ Huddling freq. | −5.71 | 0.30 | 10.92 | 0.05 | −1.7 | 0.68 |
| *E. coli* % similarity $\sim$ Aggression freq. | 3.22 | 0.05 | 7.17 | 0.02 | −0.17 | 0.97 |
| *E. coli* % similarity $\sim$ Kinship | 0.21 | 0.84 | 0.88 | 0.35 | | |

Notes.
   [a]Kinship data not analyzed for Group III on account of disproportionate representation of non-kin over close kin dyads.

**Table 4** Number of hierarchical levels and communities in the macaque DCG trees.

| DCG tree | Group I | | Group II | | Group III | |
|---|---|---|---|---|---|---|
| | H | C | H | C | H | C |
| Grooming | 5(3) | 8 | 5(3) | 8 | 5(3) | 8 |
| Huddling | 3(2) | 9 | 3(2) | 4 | 3(2) | 15 |
| Aggression | 3(2) | 4 | 4(3) | 4 | 4(3) | 5 |

Notes.
   H, Number of hierarchical levels. Values in parentheses indicate the level at which communities were identified; C, Number of communities (or clusters) identified.

established that the observed cluster memberships at all these levels were significantly different compared to membership within 1,000 randomly-generated communities ($p < 0.01$). We hence resorted to assigning communities based on cluster membership at the intermediate hierarchical level of each tree (e.g., level-2 in a 3-level tree, level-3 in a 5-level tree), to ensure both optimum sizes and numbers of communities (Table 4).

Wilcoxon rank-sum tests established that the mean similarity in *E. coli* among individuals within the same behavioral communities were consistently and significantly greater than the similarity in *E. coli* among individuals between different communities (Figs. 4A–4C; Table 5). Group III showed the greatest differences, as indicated by the highest *z* coefficients for all three types of behavioral communities. In Groups I and II, the *z* coefficients were higher for certain types of communities, specifically huddling communities in Group I and grooming in Group II. Further, randomization tests strongly supported our findings from the Wilcoxon rank-sum tests. These showed the greatest, most consistent support for contact-mediated community-wide sharing among Group III, in comparison to groups I and II in which the extent of support was contingent on the type of behavioral community. For Group III, the degree of within-community *E. coli* similarity was significantly greater than expected by chance for all three types of communities. In comparison, the results were less consistent for Group I (only the huddling network reached significance) and Group II (grooming and aggression networks, but not huddling, reached significance) (summarized in Table 5).

Out of the 18 environmental fecal samples that we processed for Group II, we confirmed and generated an *E. coli* genotypic profile from 15 samples. A Wilcoxon rank-sum test revealed that isolates from fecal swabs of individual macaques were more similar to each

A

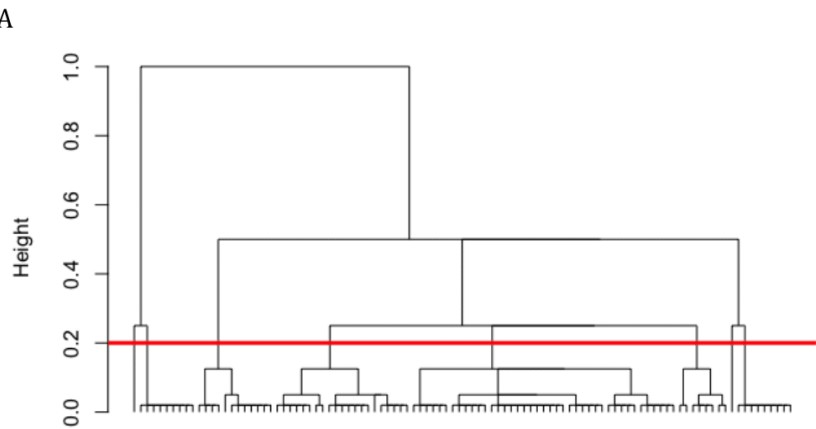

B

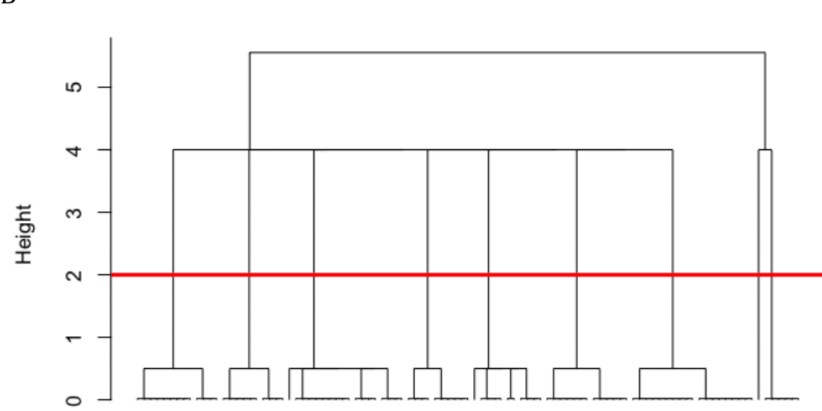

C

**Figure 3  DCG social network community structures reconstructed from (A) grooming, (B) huddling, and (A) aggressive interaction matrices for Group I macaques.** Permutation tests revealed that the assignment of cluster membership was significant at each hierarchical level of each tree ($p < 0.01$). The red line indicates the intermediate level at which community membership was assigned for the analyses.

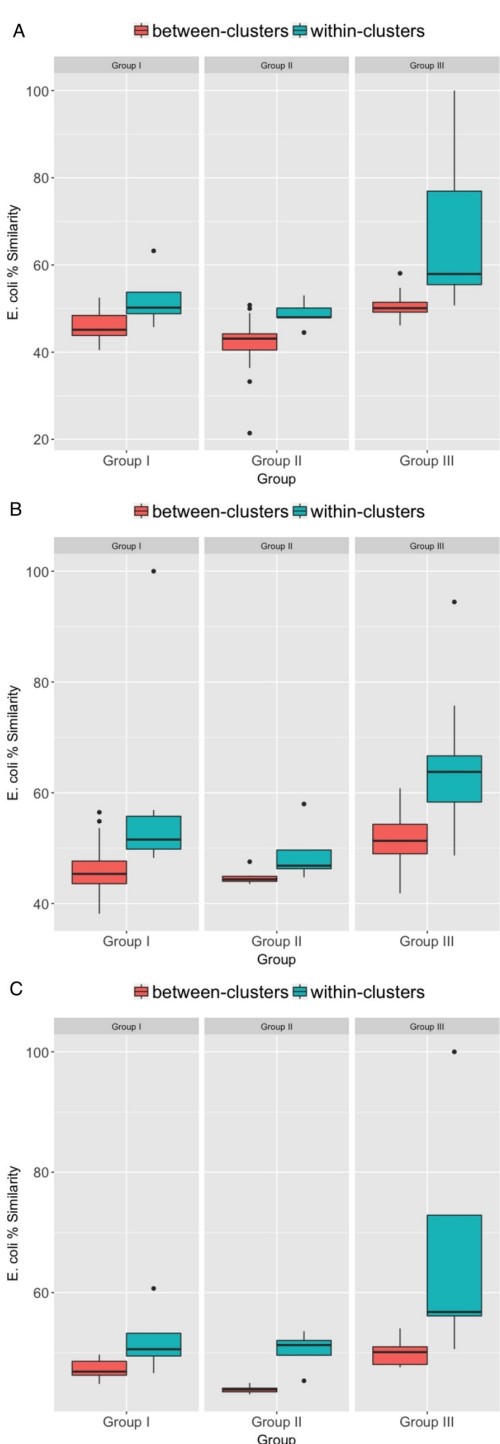

**Figure 4** Box-plots indicating differences between *E. coli* % similarity among macaques within- versus between-social network communities based on their (A) grooming, (B) huddling, and (C) aggressive interactions.

**Table 5** Results from Wilcoxon Rank-Sum tests to detect differences between *E. coli* % similarity coefficients among individuals within the same versus across different social network community clusters. Values in parentheses contain *p* values from randomization tests that compared the mean *E. coli* similarity coefficients among individuals within clusters, to means from 1,000 datasets in which cluster-membership was randomly assigned.

| DCG behavioral community | Z statistic (*p* value from randomization tests) | | |
| --- | --- | --- | --- |
| | Group I | Group II | Group III |
| Grooming | 3.82[**] (0.06) | 5.01[**] (0.05[*]) | 5.51[**] (<0.01[**]) |
| Huddling | 5.84[**] (0.05[*]) | 2.80[**] (0.25) | 9.51[**] (<0.01[**]) |
| Aggression | 2.80[**] (0.17) | 2.80[**] (0.03[*]) | 3.41[**] (0.03[*]) |

Notes.
[**]$p < 0.01$.
[*]$p < 0.05$.

other than they were to these environmental fecal isolates ($Z = 98.5$, $p < 0.01$). This confirmed that *E. coli* sharing among captive macaques is more likely to occur via social contact-mediated routes than via independent acquisition from environmental feces.

## DISCUSSION

In social systems, high frequencies of interactions with a wide range of partners may enhance contact-mediated acquisition and transmission of pathogens (reviewed in *Drewe & Perkins, 2015*; *Kappeler, Cremer & Nunn, 2015*). Here we use the microbial phylogenetics of a model commensal gut bacterium *E. coli*, along with social network reconstruction to establish a basis for social-contact mediated microbial sharing among captive groups of rhesus macaques at multiple organizational scales. Below we discuss our findings in depth, and their implications for epidemiological strategies currently in place to control the spread of infectious agents in captive and free-living animal populations.

Previous work on social taxa has revealed that the genetic similarity of commensal *E. coli* may be determined by "microbiological units" that constitute individuals from the same groups and/or metapopulations (e.g., household membership in humans and pets: *Johnson, Clabots & Kuskowski, 2008*; *Johnson et al., 2009*; group membership in wild Verreaux's sifakas: *Springer et al., 2016*). Consistent with these findings, our population genetics approach established that the genotypic variance of *E. coli* among rhesus macaques was strongly influenced by group membership, despite accounting for the effects of attributes like age, sex, and matrilineal relatedness (as in Verreaux's sifakas (*Springer et al., 2016*), but see findings on African elephants (*Chiyo et al., 2014*)). Group membership may influence heterogeneity in microbial profiles in two ways. First, environmental heterogeneity may explain bacterial heterogeneity between groups (*Chiyo et al., 2014*; *Kappeler, Cremer & Nunn, 2015*; *VanderWaal et al., 2013b*). The survival kinetics of *E. coli* in the environment may be impacted by spatiotemporal fluctuations in lighting, temperature, and/or stagnant water availability (*Habteselassie et al., 2008*; *Sinton, Hall & Braithwaite, 2007*; *Van Elsas et al., 2011*). Yet this explanation was unlikely, given that our captive study population was exposed to consistent, similarly hot and dry environmental conditions which maybe deemed unfavorable for the environmental survival of *E. coli* (see Methods for further

details). A more likely explanation is that intragroup bacterial sharing occurs via fecal-oral contact-routes. This explanation is consistent with the high variance in bacterial genetic diversity across groups, and the lack of opportunities of intergroup interactions in captivity. In other words, the finding that group membership more strongly influenced the genotypic variance of *E. coli* than attributes like age, sex, and matrilineal relatedness, establishes a premise for using network-based approaches to reveal socially-mediated, horizontal sharing of *E. coli* in this population.

Commensal *E. coli* is commonly transmitted in animals via the fecal-oral route (*Archie, Luikart & Ezenwa, 2009*). It is therefore conceivable that animals that socially interact with each other more frequently, and/or spend greater amounts of time using the same space, may be more likely to share the same *E. coli* sub-types (*Springer et al., 2016*; *VanderWaal et al., 2013b*; *VanderWaal et al., 2014a*). Within our study groups, we found no clear links between the degree of *E. coli* similarity and the frequencies of dyadic social behavioral interactions. This was in contrast to previous work on African ungulates that revealed strong, positive associations between links in microbial transmission networks based on phylogenetic relatedness, and dyadic association strengths in their social networks (*VanderWaal et al., 2013b*; *VanderWaal et al., 2014a*). Yet in each macaque group, we found that the sharing of *E. coli* was more easily discernible at the level of social communities. Specifically, individuals within well-connected clusters of grooming, huddling, and/or aggression social networks had more genotypically similar *E. coli* than they did to individuals within other clusters. In large social groups, the sub-structuring of social networks into communities may hinder, or present *bottlenecks* to the contact-mediated transmission of infectious agents (*Griffin & Nunn, 2012*; *Huang & Li, 2007*; *Nunn et al., 2011*; *Nunn et al., 2015*; *Salathe & Jones, 2010*). Here our findings reveal evidence for such "genotypic trapping" of *E. coli* within the social network communities of frequently interacting sub-groups of macaques. They therefore analogously support this social bottleneck hypothesis. Increased risk of infectious agent acquisition may have imposed selection pressures on the evolution of sociality in general, and sub-structuring of social groups into communities in particular (*Nunn, 2012*; *Nunn et al., 2015*). Further, sub-grouping may also be expected to select for shifts from individual physiological immune responses to social immunity (*Cremer, Armitage & Schmid-Hempel, 2007*; *Evans et al., 2006*). So our findings should lead naturally to future investigations that establish links between aspects of sociality (group size, social network community structure), infectious agent prevalence and transmission, and proximate indicators of individuals' immune responses (changes in physiology, gene expression), both within the CNPRC population and across a wider range of populations and taxa (*Nunn et al., 2015*).

Our detection of contact-mediated sharing at the community level but not at the dyadic level may be due to multiple host-specific factors. First, it is conceivable that in large groups of macaques where individuals may come into frequent contact with a range of partners, microbial sharing may be influenced by both direct and secondary connections. Dyadic interaction frequencies capture just the strength of individuals' direct interactions. It is now well established that in socially complex species like macaques, secondary connections in social networks maybe proxies for the occurrence of hidden/ unobserved contact-patterns

(*Balasubramaniam et al., 2016*; *Brent et al., 2010*; *Farine & Whitehead, 2015*; *MacIntosh et al., 2012*; *Makagon, McCowan & Mench, 2012*). Previous work has revealed that individuals with more primary *and* secondary connections in their social networks are (a) more likely to be infected (e.g., enteric bacteria in rhesus macaques: *Balasubramaniam et al., 2016*), and/or (b) show higher prevalence levels (e.g., nematodes in Japanese macaques: *MacIntosh et al., 2012*) of fecal-orally transmitted pathogens. Furthermore, sub-group formation and cluster membership in macaques maybe determined by both direct and secondary connections among individuals (*Sueur et al., 2011a*; *Sueur et al., 2011b*). Indeed, the DCG community structure membership, by seizing such direct and indirect connections in assigning community membership, may better capture broader-scale bacterial sharing that may go undetected at the dyadic level.

Alternatively, such anomalies between our findings at the community-level compared to the dyadic-level maybe due to methodological limitations related to microbial sampling effort. Owing to the large number of individual hosts (299 in total), our study was limited to isolating and genotyping a single *E. coli* strain from each individual macaque at the end of the behavioral data collection period. Yet in addition to a single, predominant strain, an individual may have up to 13 different strains of commensal *E. coli* (*Ahmed, Olsen & Herrero-Fresno, 2017*; *Anderson, Whitlock & Harwood, 2006*; *Bok et al., 2013*). Typing a single strain per individual can bring about anomalies in results, such as the lack of evidence for contact-mediated sharing at the dyadic level despite the detection of strong signals for sharing at higher levels of organization (here behavioral communities, groups). To capture bacterial sharing events among dyads, future work will need to conduct both sampling and comparisons of *E. coli* at multiple time-points within the behavioral sampling period, as well as more intensive sampling of four or more clones per individual within the same time-point (as in *Springer et al., 2016*; *VanderWaal et al., 2013b*).

In gregarious animals, fecal-oral microbial sharing may occur either because of social contact, or through contaminated environmental space (*Chiyo et al., 2014*; *Kappeler, Cremer & Nunn, 2015*; *Nunn et al., 2011*; *Springer et al., 2016*). So one concern was that the observed links between *E. coli* similarity and social network community structure in macaques may have been influenced by their shared space use. In general, parsing out the relative effects of spatial versus social contact on microbial sharing may be complicated by space-use being a pre-requisite (and hence a strong correlate) of social contact (*Altizer et al., 2003*; *Kappeler, Cremer & Nunn, 2015*; *Nunn et al., 2011*). In free-living animals, studies that have assessed links between intergroup or interindividual home-range overlap and microbial sharing have yielded mixed findings. For instance, *Springer et al. (2016)* found that intergroup spatial overlap, but also rates of encounters that may have involved direct social contact, were both strongly associated with *E. coli* subtype sharing in Verreaux's sifakas. In reticulated giraffes, *VanderWaal et al. (2013b)* detected no direct links between spatial overlap networks and *E. coli* sharing. Rather, they found that aspects of giraffe space-use patterns seemed to be closely linked to their social connections, which directly affected *E. coli* sharing. In comparison with these previously studied free-living animal populations, our study system of captively housed rhesus macaques presents a more spatially-constrained but socially complex context, which may be expected

to display both higher frequencies and broader repertoires of social contact behaviors (*Kaplan, 1978*; *Sade, 1972*; *Thierry, 2007*) which may facilitate microbial sharing. Further, although some strains of *E. coli* may persist in the environment for between eight-25 weeks (*Habteselassie et al., 2008*), the exposure of our study population to dry weather and low moisture content may greatly reduce the environmental survival time of gut *E. coli*, which require moist conditions (*Habteselassie et al., 2008*; *Sinton, Hall & Braithwaite, 2007*; *Van Elsas et al., 2011*). Given these system-specific conditions, it was unlikely that the fecal-oral sharing of *E. coli* strains occurs via macaques' shared space-use exposing them to environmental feces. Indeed, our finding that for Group II, *E. coli* isolated directly from macaque rectal swabs were more genotypically similar to each other than to *E. coli* isolated from environmentally deposited feces ssupports this claim.

A final potential concern was that the phylogenetic relationships of *E. coli*, in addition to horizontal sharing events, would also reflect their evolutionary relationships (*Liu et al., 2010*; *Wallace et al., 2007*). Our comparisons of fingerprint profiles rather than haplotypes limit the ability to detect, and indeed account for bacterial genetic distances that may arise due to nucleotide substitution and/or mutation events (*Archie & Ezenwa, 2011*; *Beja-Pereira et al., 2009*). That said, evolutionary change typically occurs over longer durations of time as compared to more epidemiologically relevant, short-term sharing events. So such phylogenetic signals, although present, may not be expected to mask horizontal sharing (*VanderWaal et al., 2014a*).

Our results revealed possible intergroup differences in the strength and consistency of contact-mediated *E. coli* sharing. Specifically, Group III showed both a significantly higher (than Groups I and II) mean *E. coli* similarity coefficient, as well as significantly greater within- compared to between-community similarity in *E. coli* for all three types of behavioral communities. Reasons for this may stem from variation in the groups' social stability. Specifically, Groups I and II were more socially stable than Group III, i.e., maintained consistent, stable dominance hierarchies, as evidenced by fewer reversals in the overall directions of dominance encounters across their aggression and submissive status networks (*Beisner et al., 2015*; *Chan et al., 2013*). In Group III, a comparison of these networks revealed marked inconsistencies in the direction of the relationships, which persisted until the group suffered a social collapse around 13 weeks after the data collection period (*Beisner et al., 2015*; *Chan et al., 2013*). During periods of social instability, individuals may show higher rates of uni- and bi-directional aggressive interactions (*Beisner et al., 2011*), but may also spend greater durations of time affiliating with fewer, preferred partners within their communities (*Sueur et al., 2011a*). Consistent with this, we detected more huddling (but not grooming) clusters in Group III compared to Groups I and II (Table 4). Further, previous work established that the contact-mediated acquisition of a pathogenic bacterium (*Shigella flexneri*) was also more easily discernible in Group III, compared to Groups I and II, in which stable conditions seemed to socially buffer well-connected individuals from *Shigella* infection rather than expose them to contact-mediated acquisition (*Balasubramaniam et al., 2016*). More definitive conclusions await future work that establishes links between group social stability and social contact-frequencies across multiple CNPRC macaque groups.

In conclusion, our findings establish strong links between social network community membership and the sharing of commensal *E. coli* in rhesus macaques. The population structure of *E. coli* in accordance with group membership favors an explanation of fecal-oral contact-based acquisition, most likely mediated via within-group social interactions among individuals within the same behavioral communities. Our findings have implications for both the management of captively housed animal social groups and the conservation of free-living groups and populations. Specifically, microbial sharing demonstrated by commensal *E. coli* may serve as a translational model for the acquisition and transmission of more severe, fecal-oral pathogens that are epidemiologically similar (including enteric bacteria such as pathogenic *E. coli* O157:H7, *Shigella spp.*, *Cryptosporidium spp*, some helminthes, etc.: *VanderWaal et al., 2014a*; *VanderWaal & Ezenwa, 2016*). Thus they allow for assessments of potential transmission pathways through animal groups without waiting for a clinical epidemic, or making *post hoc* conclusions about its transmission patterns after an epidemic has occurred. They also encourage epidemiological assessments to focus on multiple social or organizational scales, e.g., individual superspreaders, frequently interacting dyads, communities of preferred social partners that are likely to "trap" infectious agents, before designing targeted disease-control strategies like vaccination to check the flow of epidemics.

## ACKNOWLEDGEMENTS

We would like to thank our dedicated team from the McCowan Animal Behavior Laboratory, including A Nathman, A Barnard, T Boussina, A Vitale, E Cano, J Greco, N Sharpe, and S Seil, who participated in the behavioral data collection. We would also like to thank members of the Atwill, McCowan, and WIFSS (Western Institute for Food Safety and Security) laboratories, particularly C Bonilla, J Carabez, A Maness, R Pisano, I Wong, and C Xiao, for playing key roles in the processing of fecal samples for bacterial isolation, characterization, and fingerprinting at the School of Veterinary Medicine, UC Davis. Finally, we are grateful to Kevin Fujii for his input regarding the generation of permuted networks of varying sampling effort. The content is solely the responsibility of the authors and does not necessarily represent the official views of the National Institutes of Health.

### Funding

Research reported in this publication was supported by NICHD of the National Institutes of Health under award number R01HD068335. The funders had no role in study design, data collection and analysis, decision to publish, or preparation of the manuscript.

### Grant Disclosures

The following grant information was disclosed by the authors:
National Institutes of Health: R01HD068335.

## Competing Interests

The authors declare there are no competing interests.

## Author Contributions

- Krishna Balasubramaniam conceived and designed the experiments, performed the experiments, analyzed the data, wrote the paper, prepared figures and/or tables.
- Brianne Beisner conceived and designed the experiments, analyzed the data, wrote the paper, reviewed drafts of the paper.
- Jiahui Guan analyzed the data, prepared figures and/or tables.
- Jessica Vandeleest conceived and designed the experiments, performed the experiments, wrote the paper.
- Hsieh Fushing contributed reagents/materials/analysis tools.
- Edward Atwill and Brenda McCowan conceived and designed the experiments, contributed reagents/materials/analysis tools, reviewed drafts of the paper.

## Animal Ethics

The following information was supplied relating to ethical approvals (i.e., approving body and any reference numbers):

The protocols used for this research were approved by the UC Davis Institutional Animal Care and Use Committee (IACUC), and were in accordance with the legal requirements of the jurisdictions in which the research was conducted. UC Davis IACUC Protocol #: 18525. Office of Laboratory Animal Welfare (OLAW) Assurance Number: A3433-01.

## Data Availability

The raw datasets have been provided as Supplemental Files and is deposited at Balasubramaniam, Krishna et al. (2018), Social network community structure and the contact-mediated sharing of commensal E. coli among captive rhesus macaques (Macaca mulatta), v2, DataONE Dash, Dataset, https://dx.doi.org/10.15146/R37M2P.

## Supplemental Information

Supplemental information for this article can be found online at http://dx.doi.org/10.7717/peerj.4271#supplemental-information.

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

# PeerJ

**Evans JD, Aronstein K, Chen YP, Hetru C, Imler JL, Jiang H, Kanost M, Thompson GJ, Zou Z, Hultmark D. 2006.** Immune pathways and defence mechanisms in honey bees *Apis mellifera*. *Insect Molecular Biology* **15**:645–656 DOI 10.1111/j.1365-2583.2006.00682.x.

**Excoffier L, Smouse PE, Quattro JM. 1992.** Analysis of molecular variance inferred from metric distances among DNA haplotypes: application to human mitochondrial DNA restriction data. *Genetics* **131**:479–491.

**Farine DR, Whitehead H. 2015.** Constructing, conducting and interpreting animal social network analysis. *Journal of Animal Ecology* **84**:1144–1163 DOI 10.1111/1365-2656.12418.

**Ferrin DL, Dirks KT, Shah PP. 2006.** Direct and indirect effects of third-party relationships on interpersonal trust. *Journal of Applied Psychology* **91**:870–883 DOI 10.1037/0021-9010.91.4.870.

**Fushing H, McAssey MP. 2010.** Time, temperature, and data cloud geometry. *Physical Review E* **82**:Article 061110 DOI 10.1103/PhysRevE.82.061110.

**Fushing H, Wang H, VanderWaal K, McCowan B, Koehl P. 2013.** Multi-scale clustering by building a robust and self correcting ultrametric topology on data points. *PLOS ONE* **8(2)**:e56259 DOI 10.1371/journal.pone.0056259.

**Gibbons DE. 2004.** Friendship and advice networks in the context of changing professional values. *Administrative Science Quarterly* **49**:238–262.

**Godfrey SS, Bull CM, James R, Murray K. 2009.** Network structure and parasite transmission in a group-living lizard, the gidgee skink, Egernia stokesii. *Behavioral Ecology and Sociobiology* **63**:1045–1056 DOI 10.1007/s00265-009-0730-9.

**Goldberg T, Gillespie TR, Rwego IB, Estoff EL, Chapman CA. 2008.** Forest fragmentation as cause of bacterial transmission among nonhuman primates, humans, and livestock, Uganda. *Emerging Infectious Diseases journal* **14**:1375–1382 DOI 10.3201/eid1409.071196.

**Goldberg T, Gillespie TR, Rwego IB, Wheeler E, Estoff EL, Chapman CA. 2007.** Patterns of gastrointestinal bacterial exchange between chimpanzees and humans involved in research and tourism in western Uganda. *Biological Conservation* **135**:511–517 DOI 10.1016/j.biocon.2006.10.048.

**Goldberg T, Gillespie TR, Singer RS. 2006.** Optimization of analytical parameters for inferring relationships among escherichia coli isolates from repetitive-element PCR by maximizing correspondence with multilocus sequence typing data. *Applied and Environmental Microbiology* **72**:6049–6052 DOI 10.1128/AEM.00355-06.

**Good RC, May BD, Kawatomari T. 1969.** Enteric pathogens in monkeys. *Journal of Bacteriology* **97**:1048–1055.

**Griffin RH, Nunn CL. 2012.** Community structure and the spread of infectious disease in primate social networks. *Evolutionary Ecology* **26(4)**:779–800 DOI 10.1007/s10682-011-9526-2.

**Habteselassie M, Bischoff M, Blume E, Applegate B, Reuhs B, Brouder S, Turco RF. 2008.** Environmental controls of the fate of *Escherichia coli* in soil. *Water, Air, and Soil Pollution* **190**:143–155 DOI 10.1007/s11270-007-9587-6.

**Hamede RK, Bashford J, McCallum H, Jones M. 2009.** Contact networks in a wild Tasmanian devil (Sarcophilus harrisii) population: using social network analysis to reveal seasonal variability in social behaviour and its implications for transmission of devil facial tumour disease. *Ecology Letters* **12(11)**:1147–1157 DOI 10.1111/j.1461-0248.2009.01370.x.

**Hanneman RA, Riddle M. 2005.** *Introduction to social network methods.* Riverside: UC Riverside.

**Henzi SP, Barrett L. 1999.** The value of grooming to female primates. *Primates* **40(1)**:47–59 DOI 10.1007/BF02557701.

**Huang W, Li C. 2007.** Epidemic spreading in scale-free networks with community structure. *Journal of Statistical Mechanics* Article P01014. Epub ahead of print Jan 17 2007 DOI 10.1088/1742-5468/2007/01/P01014.

**Hubert LJ. 1987.** *Assignment methods in combinatorial data analysis.* New York: Marcel Dekker.

**Johnson JR, Clabots C, Kuskowski MA. 2008.** Multiple-host sharing, long-term persistence, and virulence of Escherichia coli clones from human and animal household members. *Journal of Clinical Microbiology* **46**:4078–4082 DOI 10.1128/JCM.00980-08.

**Johnson JR, Miller S, Johnston B, Clabots C, Debroy C. 2009.** Sharing of Escherichia coli sequence type ST131 and other multidrug-resistant and urovirulent E. coli strains among dogs and cats within a household. *Journal of Clinical Microbiology* **47**:3721–3725 DOI 10.1128/JCM.01581-09.

**Johnson S. 1967.** Hierarchical clustering schemes. *Psychometrika* **32(3)**:241–254 DOI 10.1007/BF02289588.

**Kaplan JR. 1978.** Fight interference and altruism in rhesus monkeys. *American Journal of Physical Anthropology* **49**:241–250 DOI 10.1002/ajpa.1330490212.

**Kaplan JR, Heise ER, Manuck SB, Shively CA, Cohen S, Rabin BS, Kasprowicz AL. 1991.** The relationship of agonistic and affiliative behavior patterns to cellular immune function among cynomolgus monkeys Macaca-fascicularis living in unstable social groups. *American Journal of Primatology* **25(3)**:157–174 DOI 10.1002/ajp.1350250303.

**Kappeler PM, Cremer S, Nunn CL. 2015.** Sociality and health: impacts of sociality on disease susceptibility and transmission in animal and human societies. *Philosophical Transactions of the Royal Society B: Biological Sciences* **370**:Article 20140116 DOI 10.1098/rstb.2014.0116.

**Kilonzo C, Atwill ER, Mandrell R, Garrick M, Villanueva V. 2011.** Prevalence and molecular characterization of Escherichia coli O157:H7 by multiple locus variable number tandem repeat analysis and pulsed field gel electrophoresis in three sheep farming operations in California. *Journal of Food Protection* **74**:1413–1421 DOI 10.4315/0362-028X.JFP.

**Kondo S, Hoar BR, Villanueva V, Mandrell RE, Atwill ER. 2010.** Longitudinal prevalence and molecular typing of Escherichia coli O157:H7 using multiple-locus variable-number tandem-repeats analysis and pulsed field gel electrophoresis

in a range cattle herd in California. *American Journal of Veterinary Research* **71**:1339–1347 DOI 10.2460/ajvr.71.11.1339.

**Krackhardt D. 1987.** QAP partialling as a test of spuriousness. *Social Networks* **9**:171–186 DOI 10.1016/0378-8733(87)90012-8.

**Ley RE, Lozupone CA, Hamady M, Knight R, Gordon JI. 2008.** Worlds within worlds: evolution of the vertebrate gut microbiota. *Nature Reviews Microbiology* **6(10)**:776–788 DOI 10.1038/nrmicro1978.

**Lindburg DG. 1971.** The rhesus monkey in north India: an ecological and behavioral study. In: Rosenblum LA, ed. *Primate behavior: developments in field and laboratory research*. New York: Academic Press, 1–106.

**Liu W, Li Y, Learn GH, Rudicell RS, Robertson JD, Keele BF, Ndjango JBN, Sanz CM, Morgan DB, Locatelli S, Gonder MK, Kranzusch PJ, Walsh PD, Delaporte E, Mpoudi-Ngole E, Georgiev AV, Muller MN, Shaw GM, Peeters M, Sharp PM, Rayner JC, Hahn BH. 2010.** Origin of the human malaria parasite *Plasmodium falciparum* in gorillas. *Nature* **467**:420–427 DOI 10.1038/nature09442.

**Lusseau D, Whitehead H, Gero S. 2008.** Incorporating uncertainty into the study of animal social networks. *Animal Behaviour* **75**:1809–1815 DOI 10.1016/j.anbehav.2007.10.029.

**MacIntosh AJJ, Jacobs A, Garcia C, Shimizu K, Mouri K, Huffman MA, Hernandez AD. 2012.** Monkeys in the middle: parasite transmission through a social network of a wild primate. *PLOS ONE* **7**:e51144 DOI 10.1371/journal.pone.0051144.

**Makagon MM, McCowan BJ, Mench JA. 2012.** How can social network analysis contribute to social behavior research in applied ethology? *Applied Animal Behavior Science* **138**:152–161 DOI 10.1016/j.applanim.2012.02.003.

**McCord AI, Chapman CA, Weny G, Tumukunde A, Hyeroba D, Klotz K, Mccord AI, Chapman CA, Weny G, Tumukunde A, Hyeroba D, Klotz K, Koblings AS, Mbora DNM, Cregger M, White BA, Leigh SR, Goldberg TL. 2014.** Fecal microbiomes of non-human primates in Western Uganda reveal species-specific communities largely resistant to habitat perturbation. *American Journal of Primatology* **76(4)**:347–354 DOI 10.1002/ajp.22238.

**McCowan B, Beisner B, Bliss-Moreau E, Vandeleest J, Jin J, Hannibal D, Hsieh F. 2016.** Connections matter: social networks and lifespan health in primate translational models. *Frontiers in Psychology* **27**:Article 433 DOI 10.3389/fpsyg.2016.00433.

**McCowan B, Beisner BA, Capitanio JP, Jackson ME, Cameron AN, Seil SK, Atwill ER, Hsieh F. 2011.** Network stability is a balancing act of personality, power, and conflict dynamics in rhesus macaque societies. *PLOS ONE* **6(8)**:e22350 DOI 10.1371/journal.pone.0022350.

**Newman MEJ. 2006.** Finding community structure in networks using the eigenvectors of matrices. *Physical Review E* **74**:Article 036104 DOI 10.1103/PhysRevE.74.036104.

**Nunn CL. 2012.** Primate disease ecology in comparative and theoretical perspective. *American Journal of Primatology* **74(6)**:497–509 DOI 10.1002/ajp.21986.

**Nunn CL, Jordan F, McCabe CM, Verdolin JL, Fewell JH. 2015.** Infectious disease and group size: more than just a numbers game. *Philosophical Transactions of the Royal Society B: Biological Sciences* **370**:Article 20140111 DOI 10.1098/rstb.2014.0111.

**Nunn CL, Thrall PH, Leendertz FH, Boesch C. 2011.** The spread of fecally transmitted parasites in socially structured populations. *PLOS ONE* **6**:e21677 DOI 10.1371/journal.pone.0021677.

**Otterstatter MC, Thomson JD. 2007.** Contact networks and transmission of an intestinal pathogen in bumble bee (*Bombus impatiens*) colonies. *Oecologica* **154**:411–421 DOI 10.1007/s00442-007-0834-8.

**Paradis E. 2010.** PEGAS: an R package for population genetics with an integrated modular approach. *Bioinformatics* **26**:419–420 DOI 10.1093/bioinformatics/btp696.

**Ribot EM, Fair MA, Gautom R, Cameron DN, Hunter SB, Swaminathan B, Barrett TJ. 2006.** Standardization of pulsed-field gel electrophoresis protocols for the subtyping of Escherichia coli O157: H7 Salmonella, and Shigella for PulseNet. *Foodborne Pathogens and Disease* **3(1)**:59–67 DOI 10.1089/fpd.2006.3.59.

**Rimbach R, Bisanzio D, Galvis N, Link A, Di Fiore A, Gillespie TR. 2015.** Brown spider monkeys (*Ateles hybridus*): a model for differentiating the role of social networks and physical contact on parasite transmission dynamics. *Philosophical Transactions of the Royal Society B: Biological Sciences* **370**:Article 20140110 DOI 10.1098/rstb.2014.0110.

**Romano V, Duboscq J, Sarabian C, Thomas E, Sueur C, MacIntosh AJJ. 2016.** Modeling infection transmission in primate networks to predict centrality-based risk. *American Journal of Primatology* **78**:767–779 DOI 10.1002/ajp.22542.

**Rushmore J, Bisanzio D, Gillespie TR. 2017.** Making new connections: insights from primate–parasite networks. *Trends in Parasitology* **33**:547–560 DOI 10.1016/j.pt.2017.01.013.

**Rwego IB, Isabirye-basuta G, Gillespie TR, Goldberg T. 2007.** Gastrointestinal bacterial transmission among humans, mountain gorillas, and livestock in Bwindi Impenetrable National Park, Uganda. *Conservation Biology* **22**:1600–1607 DOI 10.1111/j.1523-1739.2008.01018.x.

**Sade DS. 1972.** A longitudinal study of social behavior of rhesus monkeys. In: Tuttle R, ed. *The functional and evolutionary biology of primates.* Chicago: Aldine-Atherton, 378–398.

**Salathe M, Jones JH. 2010.** Dynamics and control of diseases in networks with community structure. *PLOS Computational Biology* **6**:e1000736 DOI 10.1371/journal.pcbi.1000736.

**Sapolsky RM, Romero LM, Munck AU. 2000.** How do glucocorticoids influence stress responses? Integrating permissive, suppressive, stimulatory, and preparative actions. *Endocrine Reviews* **21**:55–89 DOI 10.1210/edrv.21.1.0389.

**Sears HJ, Brownlee I, Uchiyama JK. 1950.** Persistence of individual strains of Escherichia coli in the intestinal tract of man. *Journal of Bacteriology* **59**:293–301.

**Sears HJ, Janes H, Saloum R, Brownlee I, Lamoreaux LF. 1956.** Persistence of individual strains of Escherichia coli in man and dog under varying conditions. *Journal of Bacteriology* **71**:370–372.

**Segerstrom SC, Miller GE. 2004.** Psychological stress and the human immune system: a meta-analytic study of 30 years of inquiry. *Psychological Bulletin* **130**:601–630 DOI 10.1037/0033-2909.130.4.601.

**Seil SK, Hannibal D, Beisner B, McCowan B. 2017.** Predictors of insubordinate aggression among captive female rhesus macaques. *American Journal of Physical Anthropology* **164**:558–573 DOI 10.1002/ajpa.23296.

**Schmid-Hempel P. 2017.** Parasites and their social hosts. *Trends in Parasitology* **33**:453–466 DOI 10.1016/j.pt.2017.01.003.

**Silk JB, Alberts SC, Altmann J. 2003.** Social bonds of female baboons enhance infant survival. *Science* **302**:1231–1234 DOI 10.1126/science.1088580.

**Sinton L, Hall C, Braithwaite R. 2007.** Sunlight inactivation of *Campylobacter jejuni* and *Salmonella enterica*, compared with *Escherichia coli*, in seawater and river water. *Journal of Water and Health* **5**:357–365 DOI 10.2166/wh.2007.031.

**Southwick CH, Siddiqi F. 2011.** India's rhesus population: protection versus conservation management. In: Gumert MD, Fuentes A, Jones-Engel L, eds. *Monkeys on the edge: ecology and management of long-tailed macaques and their interface with humans.* Cambridge: Cambridge University Press, 275–292.

**Springer A, Mellmann A, Fichtel C, Kappeler PM. 2016.** Social structure and *Escherichia coli* sharing in a group-living wild primate, Verreaux's sifaka. *BMC Ecology* **16**:6 DOI 10.1186/s12898-016-0059-y.

**Sueur C, Jacobs A, Amblard F, Petit O, King AJ. 2011a.** How can social network analysis improve the study of primate behavior? *American Journal of Primatology* **73**:703–719 DOI 10.1002/ajp.20915.

**Sueur C, Petit O, De Marco A, Jacobs AT, Watanabe K, Thierry B. 2011b.** A comparative network analysis of social style in macaques. *Animal Behaviour* **82(4)**:845–852 DOI 10.1016/j.anbehav.2011.07.020.

**Suomi SJ. 2011.** Risk, resilience, and gene-environment interplay in primates. *Journal of the Canadian Academy of Child and Adolescent Psychiatry* **20**:289–297.

**Tenaillon O, Skurnik D, Picard B, Denamur E. 2010.** The population genetics of commensal *Escherichia coli*. *Nature Reviews Microbiology* **8**:207–217 DOI 10.1038/nrmicro2298.

**Thierry B. 2007.** Unity in diversity: lessons from macaque societies. *Evolutionary Anthropology* **16**:224–238 DOI 10.1002/evan.20147.

**Uchino B. 2004.** *Social support and physical health: understanding the health consequences of relationships.* New Haven: Yale University Press.

**Uchino B. 2009.** Understanding the links between social support and physical health: a life-span perspective with emphasis on the separability of perceived and received support. *Psychological Science* **4**:236–255 DOI 10.1111/j.1745-6924.2009.01122.x.

**Van Elsas JD, Semenov AV, Costa R, Trevors JT. 2011.** Survival of *Escherichia coli* in the environment: fundamental and public health aspects. *IMSE Journal* **5**:173–183.

**Vandeleest J, Beisner B, Hannibal D, Nathman AC, Capitanio JP, Hsieh F, Atwill ER, McCowan B. 2016.** Decoupling social status and status certainty effects on health in macaques: a network approach. *PeerJ* **4**:e2394 DOI 10.7717/peerj.2394.

**VanderWaal KL, Atwill ER, Hooper S, Buckle K, McCowan B. 2013a.** Network structure and prevalence of Cryptosporidium in Belding's ground squirrels. *Behavioral Ecology and Sociobiology* **67**(12):1951–1959 DOI 10.1007/s00265-013-1602-x.

**VanderWaal KL, Atwill ER, Isbell LA, McCowan B. 2013b.** Linking social and pathogen transmission networks using microbial genetics in giraffe (*Giraffa camelopardalis*). *Journal of Animal Ecology* **83**:406–414 DOI 10.1111/1365-2656.12137.

**VanderWaal KL, Atwill ER, Isbell LA, McCowan B. 2014a.** Quantifying microbe transmission networks for wild and domestic ungulates in Kenya. *Biological Conservation* **169**:136–146 DOI 10.1016/j.biocon.2013.11.008.

**VanderWaal KL, Ezenwa VO. 2016.** Heterogeneity in pathogen transmission: mechanisms and methodology. *Functional Ecology* **30**:1606–1622 DOI 10.1111/1365-2435.12645.

**VanderWaal KL, Wang H, McCowan B, Fushing H, Isbell LA. 2014b.** Multilevel social organization and space use in reticulated giraffe (*Giraffa camelopardalis*). *Behavioral Ecology* **25**:17–26 DOI 10.1093/beheco/art061.

**Wallace RG, HoDac H, Lathrop RH, Fitch WM. 2007.** A statistical phylogeography of influenza A H5N1. *Proceedings of the National Academy of Sciences of the United States of America* **104**:4473–4478 DOI 10.1073/pnas.0700435104.

**Whitehead H, Dufault S. 1999.** Techniques for analyzing vertebrate social structure using identified individuals: review and recommendations. *Advances in the Study of Behavior* **28**:33–74 DOI 10.1016/S0065-3454(08)60215-6.

**Young C, Majolo B, Heistermann M, Schülke O, Ostner J. 2014.** Responses to social and environmental stress are attenuated by strong male bonds in wild macaques. *Proceedings of the National Academy of Sciences of the United States of America* **111**:18195–18200 DOI 10.1073/pnas.1411450111.

**Zagenczyk TJ, Gibney R, Few WT, Purvis RL. 2013.** The ties that influence: a social network analysis of prototypical employees' effects on job attitudes among coworkers. *Journal of Management Policy and Practice* **14**:27–42.