# Peer review of "Social network community structure and the contact-mediated sharing of commensal E. coli among captive rhesus macaques (Macaca mulatta)"

_PeerJ, doi:10.7717/peerj.4271_

## Round 0.1 · original submission · Major Revisions

As you will see all 3 reviewers like your study but have some comments which I would like you to address.

·

Basic reporting

The study investigates patterns of commensal E. coli strain sharing in relation to individual, social and community characteristics in 3 captive-housed group of Rhesus macaques, a social non-human primate. Commensal E. coli genetic diversity in subtypes or strains is extracted from feacal swabs and obtained through multiple cultures and DNA fingerprinting procedure. Social networks are built from observations of behavioural interactions during 6 weeks. The study concludes at a significant effect of group membership and of social community (based on social network clustering analysis) on the % of similarity between E. coli subtypes, and a lack of a significant effect of age, sex, matriline membership and dyadic interaction frequencies.
Overall, the study is well-designed and has the potential to be of major significance for animal epidemiology. (There is always a) But the writing is quite verbose; the introduction and discussion are a bit long-winded; and there are many repetitions throughout the text (and quite some typos). This makes for a lengthy read where the reader loses focus rapidly. I also find that the introduction could be punchier by delving right away into epidemiology, transmission, infection, and linked social processes, instead of too much advocating for social network analysis tools and describing what others did or did not do. An on-point straightforward presentation of the state-of-the-art and objectives of the study would be more interesting. This is also the case for the discussion. Specifically, I don’t think that nowadays we can say that “conventional epidemiological assessments of infectious agent acquisition often assume, or treat the probability of contact for every pair of individuals in a population as equal” lines 71-72 so the authors might want to jump right in the nowadays state-of-the-art and rather insist on the level of resolution microbial genetics offer for tackling our questions. Paragraph starting line 98: the authors might also want to mention that E. coli expresses enough genetic diversity to be able to build individual host genetic profile with enough between individual variability (see Craft, M. E. (2015). Infectious disease transmission and contact networks in wildlife and livestock. Philosophical Transactions of the Royal Society of London, Biological Sciences, 370(1669), 1–12. http://doi.org/10.1098/rstb.2014.0107).
Furthermore, there are some decisions made by the authors with regards to thinking or methodology that only appear clear in the discussion whereas they would have their space in the introduction or methods already (for instance, lines 480-484 could support the assumption that environmental contamination is low and so any sharing might be attributed to x,y,z instead, either in the intro or method; or lines 544-550 about group III social instability).
I am also missing a short paragraph about the significance of the study not only in terms of epidemiology and captive population health management but also on more fundamental questions such as social processes and evolution or host-parasite interaction and evolution. This would give a bit more meat to the article.
Finally, in general, the study is well referenced but some important ones are missing, for instance see above Craft 2015 as well as other reviews more accessible than a book chapter like: Godfrey, S. S. (2013). Networks and the ecology of parasite transmission: A framework for wildlife parasitology. International Journal for Parasitology. Parasites and Wildlife, 2, 235–245. http://doi.org/10.1016/j.ijppaw.2013.09.001
Rushmore, J., Bisanzio, D., & Gillespie, T. R. (2017). Making New Connections: Insights from Primate–Parasite Networks. Trends in Parasitology, xx, 1–14. http://doi.org/10.1016/j.pt.2017.01.013
Schmid-Hempel, P. (2017). Parasites and their social hosts. Trends in Parasitology, 33(6), 453–466. http://doi.org/10.1016/j.pt.2017.01.003
Vanderwaal, K. L., & Ezenwa, V. O. (2016). Heterogeneity in pathogen transmission: mechanisms and methodology. Functional Ecology, 30, 1606–1622. http://doi.org/10.1111/1365-2435.12645

Experimental design

The experimental design is very good and it is fantastic to be able to have such high resolution data. The authors give all the material for one to reproduce their results if one fancies. However, I feel that there are some missing bits and pieces in the analysis. Although the authors argue against not including those bits and pieces, I feel they would complete the story. One concerns including proximity or some kind of space sharing/use network as well as purely contact networks. I know that proximity and grooming are very often well-correlated but although the authors state several times that conditions (sunny and dry) are unfavorable for E. coli survival in the environment and that they did not find E.coli is tap water, they also state that E.coli can still survive up to 25 weeks. The fact that the monkeys are “forced” into proximity, a fact that the authors also highlight in the discussion, might indeed “expos[e] individuals more to environmentally deposited feces, [and] also increases the likelihood of social contact among animals that share the same space” lines 588-589. At least, I missed some more explicit discussion of this point.
A second one relates to the AMOVA. I understand the logic of conducting hierarchical modelling when one attempts to explain a pattern or phenomenon without assumptions or clear predictions. But here, 1/ there are clear predictions, and 2/ we know that factors influence each other, such that age and sex similarities and kinship influences grooming frequencies for instance which in turn could influence genetic similarity %. I am just wondering whether building a more integrative model with all factors at once would be possible? The separate testing of the effect of group membership, kinship, dyadic frequencies of interactions and individual characteristics leaves a feeling of incompleteness somehow. I am not sure but maybe a more classic generalized linear mixed model (with permutations) or Bayesian regression modelling may be able to sort out the variance linked to each factor at once?

Validity of the findings

Pending some methodological precisions and discussions, I think the findings will be robust and conclusions well linked to the questions. As said before, I also find the discussion in need of more straightforwardness. For instance, the authors spend 16 lines summarizing their results, lines 443-459, then another 6 lines shortly after, lines 464-470, repeating parts of those results before entering the actual discussion.
And of a broader picture viewing as well.

Reviewer 2 ·

Basic reporting

overall great! Although there is a lot of unnecessary pointing (e.g. see below). See general comments to the authors.

Experimental design

Excellent. The only issue is the fact that a single isolate was collected from an individual requiring cation in the interpretation of results. Some statistical tests are inappropriate for the kind of data. Again look at general comments to the author.

Validity of the findings

All findings are valid. However, caution is required in the interpretation on discordant findings. See general comments to the authors below.

Additional comments

The manuscript on social community structure and contact mediated sharing of E. coli in captive maca presents data indicating the value of social interactions as a proxy for inference of the transmission of infectious agents in group living organisms. The manuscript is worth publication in PEERJ after these issues have been addressed.
1. Some statistical tests are inappropriate and needs redoing. Kolmogorov-Smirnov tests are compare the similarity in distributions and are not appropriate for comparing differences in percent similarity in E. coli between and within social clusters. A permutation / randomization test whereby percent E. coli similarity observed within DCG communities are compared with similarity obtained after shuffling individual across DCG and calculating the percent similarity between random pairs to generate the expected sharing assuming there is no influence of DCG on percent E. coli similarity among dyads.
2. The authors make a claim that social processes are important in E. coli sharing BUT do not fully and comprehensively address the issues of non-concordant results between dyadic and group level social processes in the discussion or methods section of the manuscript. Because a single individual animal can have as many as 13 E. coli strains (see refs 1-3 below). Typing a single strain per individual can bring about anomalies in results observed in this paper such as similarity at the group level or some aggregated level such as behavioral DCGs but no signal is detectable at the dyadic level. To obtain a detectable signal of dyadic interactions a single clone in a species where an individual may have several clones co-occurring will require repeated temporal sampling as the authors allude or intensive sampling more than four or more isolates to capture the full diversity within individuals and the latter is more powerful for E coli sharing and the former for tracking transmission. This should come out clearly in the discussion of discrepancy of results

3. To make a solid claim on social processes in E. coli sharing, the authors need to run their data using hierarchical analysis of molecular variance (AMOVA) incorporating group and behavioral DCG social as to partial variance between group level effects which are environmental and those that are social (behavioral DCG)

4. Specific comments. Please see attached copy with comments
Methods: Non parametric tests do not solve issues of interdependence but issues non-normality issues in data. You need tests that take into account such dependence.
References
1. Ahmed S, Olsen JE, Herrero-Fresno A (2017) The genetic diversity of commensal Escherichia coli strains isolated from non-antimicrobial treated pigs varies according to age group. PLOS ONE 12(5): e0178623.https://doi.org/10.1371/journal.pone.0178623
2. Anderson, M.A., Whitlock, J.E. and Harwood, V.J., (2006). Diversity and distribution of Escherichia coli genotypes and antibiotic resistance phenotypes in feces of humans, cattle, and horses. Applied and Environmental Microbiology, 72(11), pp.6914-6922.
3. Bok, E., Mazurek, J., Pusz, P.A.W.E.Ł., Stosik, M.I.C.H.A.Ł. and Baldy-Chudzik, K., (2013). Age as a factor influencing diversity of commensal E. coli microflora in pigs. Pol J Microbiol, 62(2), pp.165-71.

Annotated reviews are not available for download in order to protect the identity of reviewers who chose to remain anonymous.

Reviewer 3 ·

Basic reporting

No comment.

Experimental design

See general comments below.

Validity of the findings

See comments below.

Additional comments

This paper tests social predictors of E. coli strain sharing in three groups of rhesus macaques. It adds to the growing literature on the consequences of social interactions on microbial sharing/transmission, and is distinguished by a relatively large sample size and degree of environmental control. The main finding is that higher level social interactions, captured through “clusters” of socially interacting individuals, appear to better predict E. coli strain sharing than dyadic interactions. This result is likely to suggest new analysis approaches to researchers interested in this topic. I have a number of suggestions that I hope will be of use in any revision process.

Major:
1) The finding that DCG clusters are a better predictor of strain sharing than dyadic interactions is interesting, but it is unclear whether it arises as a consequence of biology versus technical artifact. With limited behavioral sampling (6 weeks per social group, mostly collected through scan sampling), dyadic estimates of grooming, huddling, and aggression rates may simply be less accurate than cluster/group-level membership. Can you show (e.g., using subsampling) that the amount of observational data collected produces asymptotically stable estimates for the three types of dyadic interactions you consider, compared to inference of cluster membership in the DCG approach? Are dyadic grooming, huddling, and aggression correlated? (If so, would combining these measures into some sort of sociality index produce a more stable measure of contact?)

2) K-S tests are used to compare E. coli similarity across communities. K-S tests are quite sensitive, since they test for differences in the cdf of two distributions, not differences in means—which I think is what you are most interested in knowing. If a non-parametric test is needed, why not use a Wilcoxon rank-sum test instead? Also, I could not tell if these tests were one-tailed (as suggested by the motivating hypothesis) or two-tailed.

3) Estimates of kinship are based on either matriline membership or a coarse binarization into close versus distant kin. How is a “matriline” defined? (how many generations back does it go?) Why binarize estimates of kinship? In the current scheme, distant cousins are treated as equivalent to full sibs; this approach seems less likely to detect effects of relatedness on strain-sharing, if they exist, than using pedigree or genetic marker-based point estimates.

4) Can you explain what DCG levels “mean” from a biological perspective? Do they tend to capture kin groups or sets of animals with close social bonds (c.f. Silk?)

Minor
-A single E. coli strain was isolated from each animal. Did you confirm that animals only carry a single E. coli strain? If they do carry multiple strains, can you explain how it might affect your results?
-line 27/28, 559, 563: “in the absence of phenomena like social buffering” What phenomenon are you referring to, and what’s its relevance to microbial sharing?
-line 132: “largely, these studies have defined…pairs of individuals from whom identical E. coli subtypes were isolated” Tung et al (and also other studies, e.g., Moeller et al in wild chimpanzees) actually use community composition (beta diversity) instead, or in addition to, sharing of individual OTUs. These studies have not analyzed E. coli isolates.
-The sample appears to contain adults only. Any ramifications of leaving out infants and juveniles as potential links in the social networks and/or routes of transmission?
-line 414: “dyads with an E. coli similarity >70%”—does that mean 70% of bands were shared, or 70% of 0/1 values for bands? What does it mean that this threshold is a “reliable indicator of pulsotype cluster membership and similarity?”
-Figure 2: I would find it helpful to see E. coli similarity levels for between group pairs to place the within group pair results in context

---

## Round 0.2 · accepted · Accept

You have dealt with all the comments.

·

Basic reporting

No further comment

Experimental design

No further comment

Validity of the findings

No further comment

Additional comments

Great study. No further comment.

Reviewer 2 ·

Basic reporting

clear and unambiguous

Experimental design

Research questions are well defined

Validity of the findings

Statistical analysis is robust and the authors addressed the comments from previous revisions

Additional comments

None